computer modelling and simulation/computational biology

pedestrian dynamics, COVID-19, aeroplane boarding

**Author for correspondence:**
A. Srinivasan
e-mail: asrinivasan@uwf.edu

# From bad to worse: airline boarding changes in response to COVID-19

T. Islam[1], M. Sadeghi Lahijani[2], A. Srinivasan[1], S. Namilae[3], A. Mubayi[4] and M. Scotch[4]

[1]Department of Computer Science, University of West Florida, Pensacola, FL, USA
[2]Department of Computer Science, Florida State University, Tallahassee, FL, USA
[3]Department of Aerospace Engineering, Embry-Riddle Aeronautical University, Daytona Beach, FL, USA
[4]Arizona State University, Tempe, AZ, USA

AS, 0000-0003-0408-2886; SN, 0000-0003-1487-9573;
AM, 0000-0003-3936-3055; MS, 0000-0001-5100-9724

Airlines have introduced a back-to-front boarding process in response to the COVID-19 pandemic. It is motivated by the desire to reduce passengers' likelihood of passing close to seated passengers when they take their seats. However, our prior work on the risk of Ebola spread in aeroplanes suggested that the driving force for increased exposure to infection transmission risk is the clustering of passengers while waiting for others to stow their luggage and take their seats. In this work, we examine whether the new boarding processes lead to increased or decreased risk of infection spread. We also study the reasons behind the risk differences associated with different boarding processes. We accomplish this by simulating the new boarding processes using pedestrian dynamics and compare them against alternatives. Our results show that back-to-front boarding roughly doubles the infection exposure compared with random boarding. It also increases exposure by around 50% compared to a typical boarding process prior to the outbreak of COVID-19. While keeping middle seats empty yields a substantial reduction in exposure, our results show that the different boarding processes have similar relative strengths in this case as with middle seats occupied. We show that the increased exposure arises from the proximity between passengers moving in the aisle and while seated. Such exposure can be reduced significantly by prohibiting the use of overhead bins to stow luggage. Our results suggest that the new boarding procedures increase the risk of exposure to COVID-19 compared with prior ones and are substantially worse than a random boarding process.

# 1. Introduction

Outbreaks of several serious diseases, such as SARS, influenza, measles and tuberculosis, have occurred during air travel [1–5]. Concern over such spread of infection led to a sharp decline in air travel during the current COVID-19 outbreak, with passenger traffic estimated to be 44–59% below normal in 2020 [6]. Airlines have responded to this with several procedural changes. Two key changes include lowering occupancy by keeping the middle seat empty and changes to boarding procedures [7,8].

For example, Delta Airlines has introduced a boarding procedure where it boards the plane starting from the last row, although business class still boards first [7]. United Airlines uses a back-to-front by rows boarding process with business boarding last typically [8]. The trend towards back-to-front boarding by rows is motivated by the desire to reduce the likelihood of passengers passing others when they take their seats [7]. Although such a boarding process is known to be slower than alternative processes [9], a reduction in social proximity would lead to a reduced risk of exposure to viruses, and thus reduced risk of an infection outbreak inflight.

However, our past results suggest that such a process may not reduce social proximity [10]. In prior work on the risk of Ebola and SARS spread during boarding aeroplanes, we observed that passengers' clustering while waiting for others to stow their luggage had a significant impact on increasing infection risk. Such clustering tended to increase with more zones that were in contiguous sections of the aeroplane. Back-to-front boarding by rows is equivalent to one zone per row and can thus be expected to yield increased clustering and exposure to the virus. There is a need to understand better the impact of the new boarding processes on exposure to the virus.

In this work, we simulate several boarding processes and evaluate their impact on social proximity during boarding. We also study the dominant mechanisms through which they generate social proximity. Our results show that while a back-to-front boarding does indeed reduce exposure of seated passengers to those who are walking past them toward their seats, it increases proximity between pairs of seated passengers and pairs of passengers in the aisle. The net impact is to increase exposure by around 100% compared with random boarding. It also increases exposure by approximately 50% compared with a typical boarding process prior to the COVID-19 outbreak. While keeping middle seats empty yields a substantial reduction in exposure, the boarding processes show the same relative trends.

Our results suggest that airlines would have a lower infection risk from their pre-COVID-19 boarding process. They can reduce the risk even further by using a random boarding process. Prohibiting the use of overhead bins to stow luggage and boarding window seats before aisle seats can also yield a substantial reduction in exposure.

# 2. Methods

Pedestrian dynamics models simulate the walking movement of humans, which can be used to determine contact patterns. A variety of models have been developed [11,12]. Social dynamics models [13,14] are the most popular and are motivated by the idea of molecular dynamics. They treat each pedestrian as analogous to an atom. The force $f_i$ acting on the $i$th pedestrian is defined as follows.

$$f_i = m_i \frac{\mathrm{d}v_i}{\mathrm{d}t} = \frac{m_i}{\tau}(v_{0i}(t) - v_i(t)) + \sum_{i \neq j} f_{ij}, \qquad (2.1)$$

where $v_i(t)$ is the actual velocity at time $t$ of a pedestrian with mass $m_i$ whose desired velocity is $v_{0i}(t)$. The momentum generated by a pedestrian's desire to move towards a goal (governed by the difference between desired and actual velocity and a time constant $\tau$) results in self-propulsion towards that goal. This is countered by a pairwise repulsion force $f_{ij}(t)$ due to interactions with other pedestrians and fixed surfaces, such as walls. The numerical solution of the above ODE for each pedestrian generates a trajectory for each pedestrian.

The repulsion term $f_{ij}$ is modelled through empirically determined force fields, often inspired by molecular dynamics [10]. Social dynamics models differ in the manner in which $f_{ij}$ is defined. *Additional behavioural features need to be explicitly coded.* For example, passengers spend a certain amount of time stowing their luggage, and a passenger going to the window seat experiences a delay if the aisle seat is occupied.

We had earlier developed the self-propelled entity dynamics (SPED) model to simulate boarding and deplaning with a specific form of $f_{ij}$ based on the Lennard–Jones potential from molecular dynamics [10].

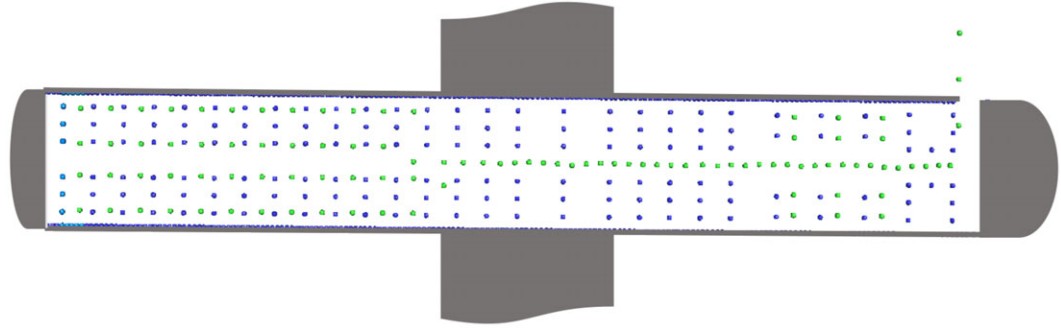

**Figure 1.** Snapshot from a simulation that boards the business class first, followed by the economy class in a back-to-front manner. Green dots represent passengers, and blue dots fixed objects, such as seats and walls.

We recently proposed the constrained linear movement (CALM) model designed for narrow passages and evaluated its correctness in boarding and deplaning (figure 1) [15]. The CALM mod*el defines $f_{ij}$ by setting $f_{ik}$ to 0 for all k except the nearest neighbour in the direction of motion. If $d_i$ is the distance to the nearest neighbour l in the direction of motion, then $f_{il}$ is given by equation* (2.2). The solution of equations (2.1) and (2.2) yields each passenger's position as a function of time. Certain behavioural features of passenger movement, such as the time to stow luggage, are accounted for directly in the simulation code, with their parameters presented in §3.1. The passenger trajectories are then used to determine the extent of social proximity between passengers.

$$f_{il} = \frac{m_i(\beta - 1)v_{0i}}{\tau}, \quad \beta = c - e^{-a(d_i - b)}, \tag{2.2}$$

where $a$, $b$ and $c$ are constants determined as $a = 2.11$, $b = 0.366$ and $c = 0.966$ in [15]. The mass term cancels out from both sides of equation (2.1) when we substitute equation (2.2) into equation (2.1), and $\tau$ is assigned the value 0.4 s.

The CALM model is around 60 times faster than the SPED model. This computational efficiency leads to its effectiveness in the analysis of different boarding procedures for the following reason. Precise prediction is difficult because human movement behaviour has inherent uncertainties and variations [16,17], compounded by insufficient data during a new epidemic [18]. We address this challenge by parametrizing the sources of uncertainty, such as passenger walking speed, and then evaluate social proximity under various possible scenarios. This parameter space is large, and it is computationally demanding even on a parallel computer. The computationally efficient CALM model enables us to analyse this large parameter space relatively fast. We use a low discrepancy parameter sweep to reduce further the number of simulations needed [19].[1]

# 3. Results

## 3.1. Simulation set-up

We perform a parameter sweep, varying the following parameters, to generate 16 000 scenarios of possible passenger movement patterns during each boarding process, with choices based on [15]. The desired speed of a passenger in the absence of anyone nearby, $v_{0i}$, has a range between 1.1 m s$^{-1}$ and 1.3 m s$^{-1}$. Parameters for behavioural features are selected as follows. Passengers are tracked from when they are just outside the plane door. They start entering the plane once the passenger in front of them is a certain distance away, which is specified by a *line-distance threshold* that ranges from 0.5 m to 1.6 m. Passengers decrease their speed after entering the plane when they turn toward the aisle. This is simulated by multiplying their speed by an *intersection-speed coefficient*, ranging between 0.2 and 0.8,

---

[1]A low discrepancy sequence fills a space uniformly. A low discrepancy parameter sweep uses such a sequence to choose values of parameters that are uniformly spread through the space. A pseudorandom choice of points, on the other hand, can yield some clustering in the set of parameters chosen. A conventional lattice-based parameter sweep, where values in each dimension are equally distributed, does not cover a space as uniformly either. For example, a lattice-based parameter sweep with two variables might choose 100 points along each direction, leading to 10 000 parameter combinations being simulated. A two-dimensional low discrepancy sequence with 10 000 points would cover such a space much more uniformly. This can result in one to three orders of magnitude fewer points being needed in a low discrepancy parameter sweep [19].

when they are within a distance *intersection-distance threshold* of the aisle. The latter parameter varies between 0.4 m and 0.6 m.[2]

*Luggage stowing time* ranges from 8 s to 14 s.[3] *Seat conflict time* varies from 5 s to 8 s [20];[3] a seat conflict arises when a seated passenger needs to give way to a passenger who will sit closer to the window, for example, when a seated aisle passenger needs to let a window passenger through. After stowing the luggage, passengers move into their seats at a fraction of the normal speed. This is simulated by multiplying their speed by the *towards-seat-speed coefficient*, which has a range from 0.2 to 0.6. We also examine situations without luggage stow time and seat conflict time. These correspond, for example, to policies that prohibit the use of the overhead bin to stow luggage and boarding window seats before the aisle seats, respectively.

Equation (2.1) is solved numerically using an Euler solver with a time-step size of 0.005 s. We output results every 250 time-steps so that each output represents 1.25 s of real time. For each simulation, we perform the following analysis. At each time step, we measure social proximity through contacts between pairs of passengers, where passengers whose distance to each other is less than a specified threshold are considered to be in contact. Each such pair yields 2.5 person-seconds of contact, indicating that two persons have been in social proximity for 1.25 s each. Thus, our metric computes the average total exposure per person.

Such total exposure is used in dose–response models to compute infection risk [21,22]. A dose–response relationship is not known for COVID-19. However, such relationships are monotonic, and thus increased exposure leads to increased risk. Note that the definition of contacts for infection modelling purposes may differ from that for contact tracing purposes because the latter needs to balance the risk of infection with the effort of contact tracing [23]. In particular, the definition for contact tracing with COVID-19 can miss infections [23]. In fact, known outbreaks have occurred [24] outside the contact threshold limit specified by the Centers for Disease Control and Prevention [25].

## 3.2. Impact of boarding procedures

We simulated several boarding procedures. We present the results of the following four. The results are for parameter sweeps that generated 16 000 scenarios, using various choices for the parameter values as specified in §3.1.

1. *1-Zone*: Passengers board in random order with no preference given to business class passengers.
2. *6-Zones business-first*: Business class passengers board first, and the rest of the plane is divided into six contiguous zones of a few rows each. These zones are boarded starting from the back-most zone and moving forward. Within each zone, boarding is random.
3. *Back-to-front*: This simulates row-wise boarding, starting with the last row. No preference is provided to business class passengers. This is similar to a recent United Airlines' boarding process.
4. *Back-to-front business-first*: Business class boards first, and then the economy class boards, in a back to front manner. This is similar to Delta Airlines' new boarding process.

We show below that the random 1-Zone boarding process is the most effective among those that we studied in reducing social proximity. It differs from the random process employed in airlines such as Southwest. In the latter, passengers may sit anywhere they wish to, and their seating preference may not yield a random order. In our simulations, passengers in this process are assigned specific seats, but the boarding order is random. The 6-Zones business-first procedure is meant to approximate a common procedure before COVID-19, where the business class boards first, and then people board in a large number of zones, with random order within a zone. Both the back-to-front procedures emulate new boarding procedures that have been adopted by different airlines.

Some airlines are keeping the middle seat empty while others are not [26]. We simulate both situations. We compare these policies with and without seat conflicts. We use a 1.82 m threshold for contacts, corresponding to the 6-feet threshold advocated by public health agencies. Figures 2 and 3 show that 1-Zone performs best, 6-Zones next best, and both the new procedures fare worst, yielding

---

[2]We use a subset of the range in [15] for this parameter. The reason is that [15] simulated four aeroplanes, ranging from 50 seats to over 200 seats. The current paper deals with only one aeroplane and does not need to consider the lowest or highest ends of the range.

[3]This value is generated for each passenger using a random number generator, rather than being a simulation parameter generated using a low discrepancy sequence.

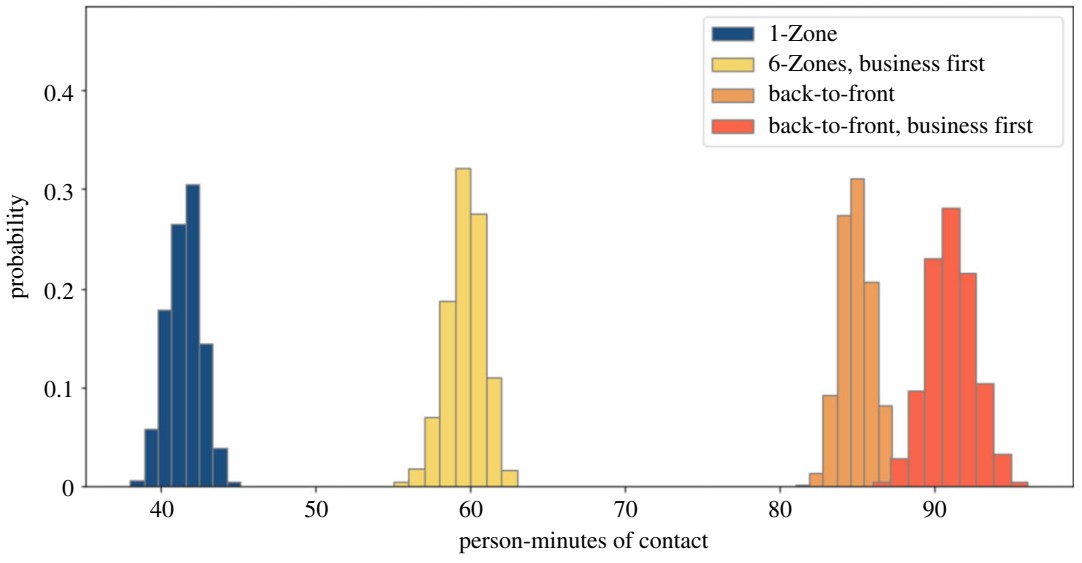

**Figure 2.** Social proximity with overhead bin use permitted, middle seats occupied, and a 1.818 m contact threshold without seat conflict.

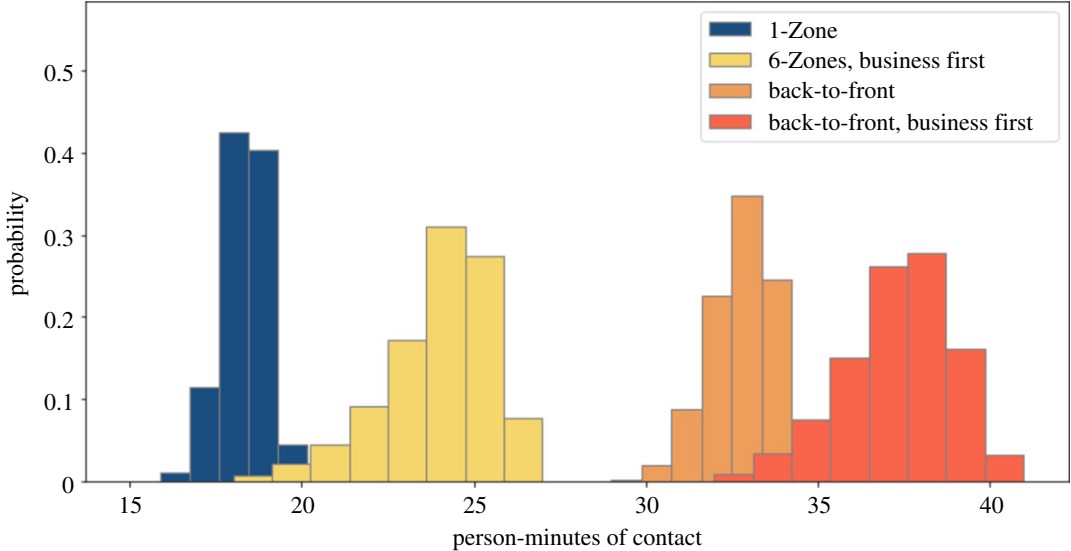

**Figure 3.** Social proximity with overhead bin use permitted, middle seats unoccupied, and a 1.818 m contact threshold without seat conflict.

roughly double the contacts of the best and 50% more than the procedure that they replaced. On the other hand, the policy of keeping middle seats empty, which has been introduced by a few airlines, delivers a substantial reduction in social proximity. Figures 4 and 5 show a similar trend when including seat conflicts, with the actual exposure being higher than the case without seat conflicts, as expected.

## 3.3. Impact of prohibiting overhead bin use

We simulated the impact of prohibiting overhead bin use by setting the luggage stow time parameter to zero. We present results for the four boarding procedures studied above, with and without seat conflicts. Figures 6 and 7 show that, without seat conflicts, there is a substantial overlap in the magnitude of exposures, unlike in the cases studied earlier. Table 1 later indicates that the average exposures are similar when middle seats are unoccupied, along with a prohibition on overhead bin use and absence of seat conflicts. Back-to-front boarding is substantially worse than the alternatives in all other cases.

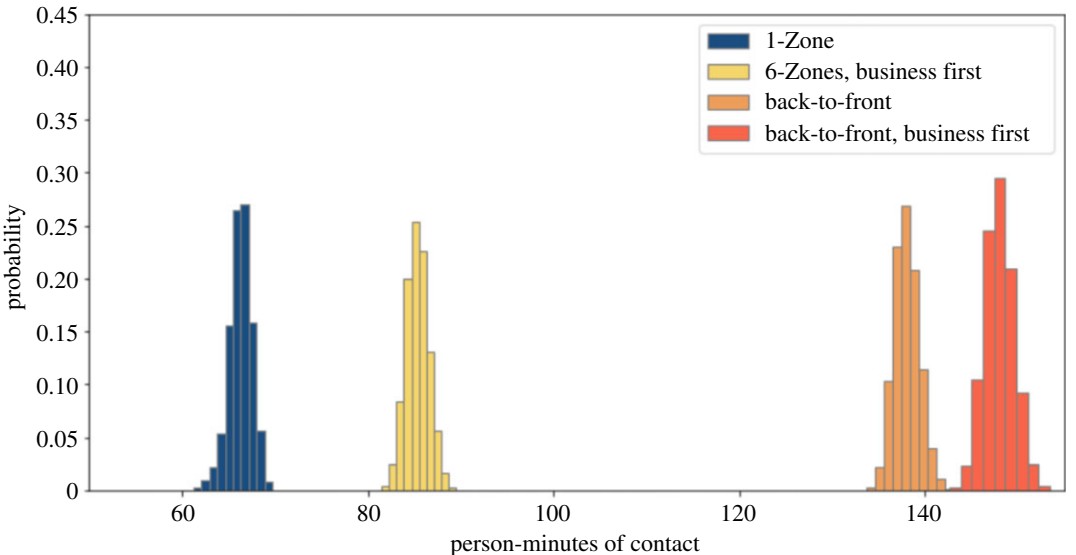

**Figure 4.** Social proximity with overhead bin use permitted, middle seats occupied, and a 1.818 m contact threshold with seat conflict.

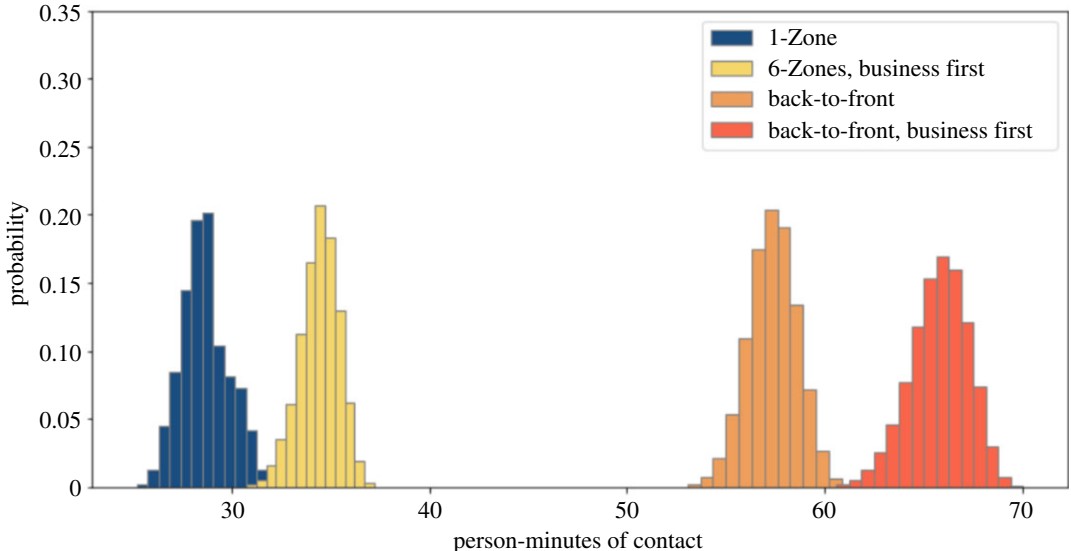

**Figure 5.** Social proximity with conflict with overhead bin use permitted, middle seats unoccupied, and a 1.818 m contact threshold with seat conflict.

Figures 8 and 9 show that the back-to-front boarding procedures perform worse than the other options when there are seat conflicts, although the resulting exposure is not as high as when overhead bin use is permitted.

## 4. Discussion

We now examine how contacts are generated. In earlier work, we had noticed an increase in social proximity for a large number of zones by analysing the trajectories of specific simulation results [10]. Here, we systematically measure proximity by determining the number of contacts arising from each of the following situations.

1. Contacts between a pair of passengers, both of whom are seated.
2. Contacts between a pair of passengers, both of whom are in the aisle.
3. Contacts between passengers, one of whom is seated and the other in the aisle.

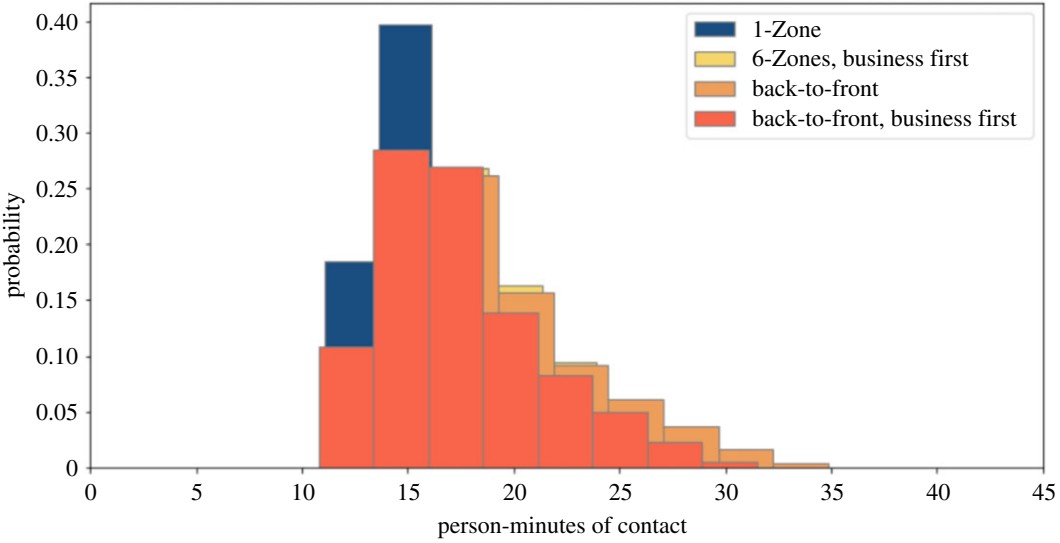

**Figure 6.** Social proximity with overhead bin use prohibited, middle seats occupied, and a 1.818 m contact threshold without seat conflict.

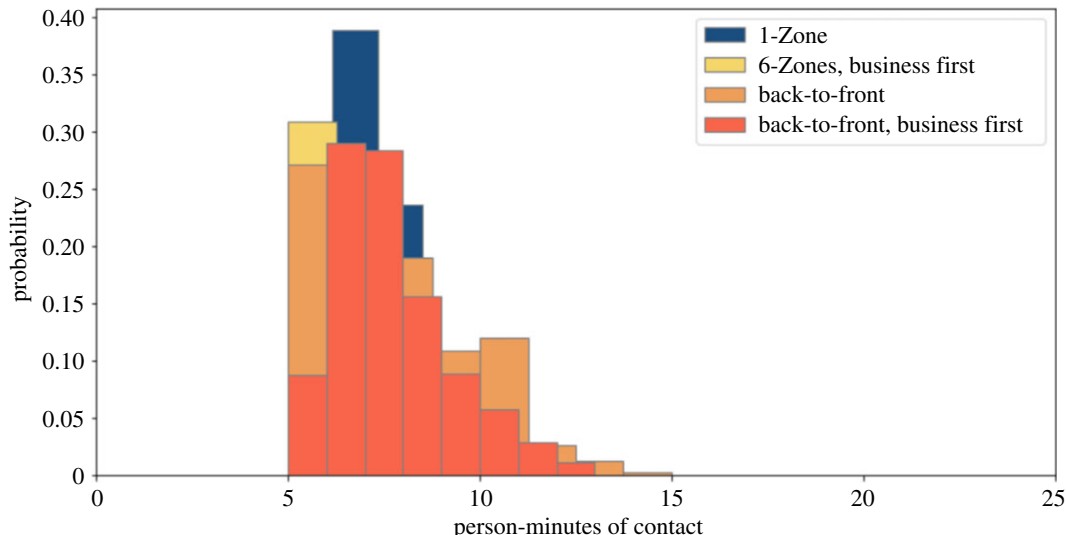

**Figure 7.** Social proximity with overhead bin use prohibited, middle seats unoccupied, and a 1.818 m contact threshold without seat conflict.

The new boarding procedures adopted by airlines try to lower the value of the last category. Our previous work hypothesizes that this would not be beneficial due to an increase in the contacts from the second category. We look at one specific typical simulation from each boarding process and analyse its contact patterns to generate insight into the mechanisms yielding social proximity.

Figure 10 shows that the back-to-front procedures do, indeed, reduce contacts between seated passengers and those moving in the aisle. However, this is not the primary generator of social proximity. An increase in contacts between passengers who are both in the aisle or both seated makes the new procedures substantially worse than the previous one and even more when compared with using a single zone. These results also show that while the hypothesized second factor is sufficient to deliver better social distancing with fewer zones, it is not the primary driver. The most important factor that yields better social distancing with fewer zones arises from the contacts generated between seated passengers. With more zones, passengers remain seated close to each other during the boarding process, while with one zone, they are distributed around the plane.

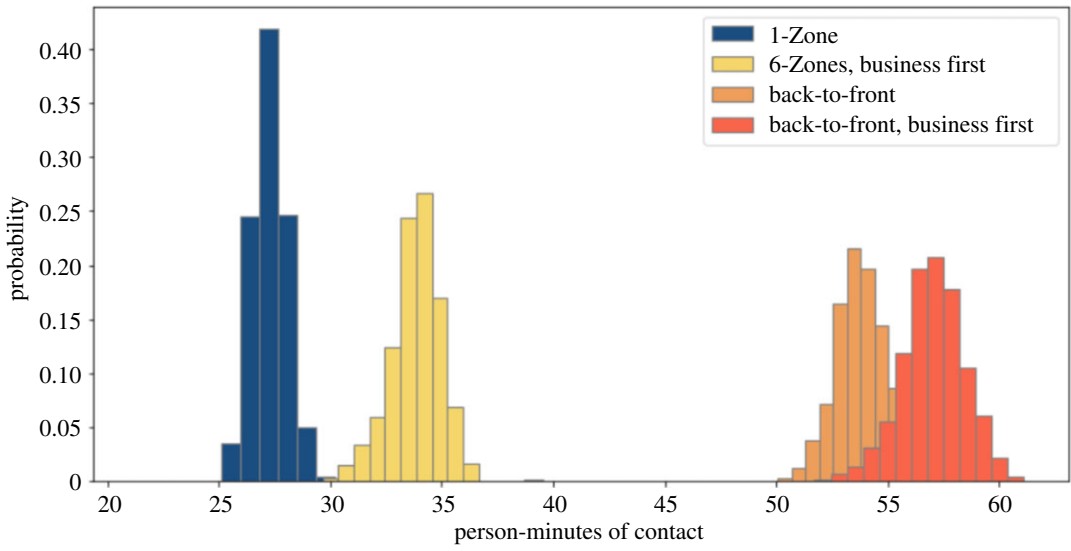

**Figure 8.** Social proximity with overhead bin use prohibited, middle seats occupied, and a 1.818 m contact threshold with seat conflict.

**Table 1.** Average exposure time per passenger (minutes).

| boarding procedures | middle seat occupied | | | | middle seat unoccupied | | | |
| --- | --- | --- | --- | --- | --- | --- | --- | --- |
| | overhead bins permitted | | overhead bins prohibited | | overhead bins permitted | | overhead bins prohibited | |
| | without seat conflict | with seat conflict | without seat conflict | with seat conflict | without seat conflict | with seat conflict | without seat conflict | with seat conflict |
| 1-Zone | 43.0 | 66.2 | 16.8 | 27.3 | 18.3 | 28.6 | 7.5 | 12.3 |
| 6-Zones, business first | 60.0 | 85.3 | 22.8 | 33.8 | 24.0 | 34.4 | 7.5 | 12.4 |
| back-to-front | 86.1 | 138.0 | 31.5 | 53.8 | 33.2 | 57.6 | 7.7 | 21.0 |
| back-to-front, business first | 92.1 | 148.0 | 30.4 | 57.0 | 37.5 | 65.8 | 7.2 | 23.3 |

Some airlines have introduced a new policy of reducing occupancy by keeping middle seats empty. While the motivation for this is to yield social distancing during the flight, we wish to see its impact on the boarding process. In particular, are the relative differences between boarding processes preserved with this seating policy? Figure 10 shows that the relative differences are maintained and that 1-Zone remains around twice as effective as the new boarding procedures. We also note that the new seating policy produces a substantial reduction in social proximity during the boarding process.

Figure 11 examines the dominant mechanisms with seat conflicts. The exposure in each boarding procedure increases as expected, with the same relative trends. The random process is even a little more advantageous than without seat conflicts. This improvement arises from two primary factors. First, with seat conflicts, the boarding process takes longer. This increases the exposure between two seated passengers, which is a particular weakness of the back-to-front boarding procedures. In addition, seat conflicts also increase the clustering in the aisle, to which these boarding procedures are again more vulnerable.

Figures 12 and 13 present the results of a similar analysis, but with a prohibition on the use of overhead bins. The exposure in each boarding procedure decreases as expected. The relative

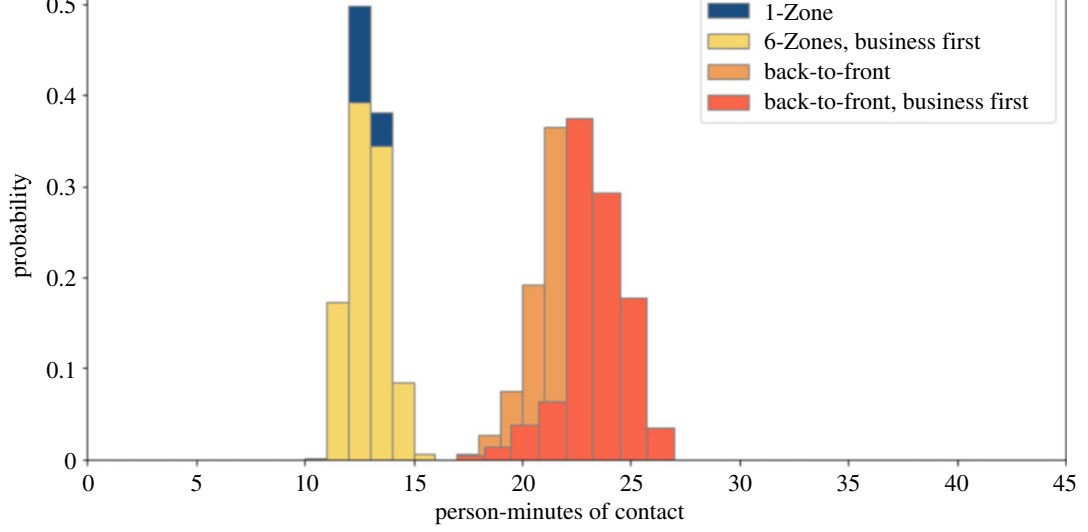

**Figure 9.** Social proximity with overhead bin use prohibited, middle seats unoccupied, and a 1.818 m contact threshold with seat conflict.

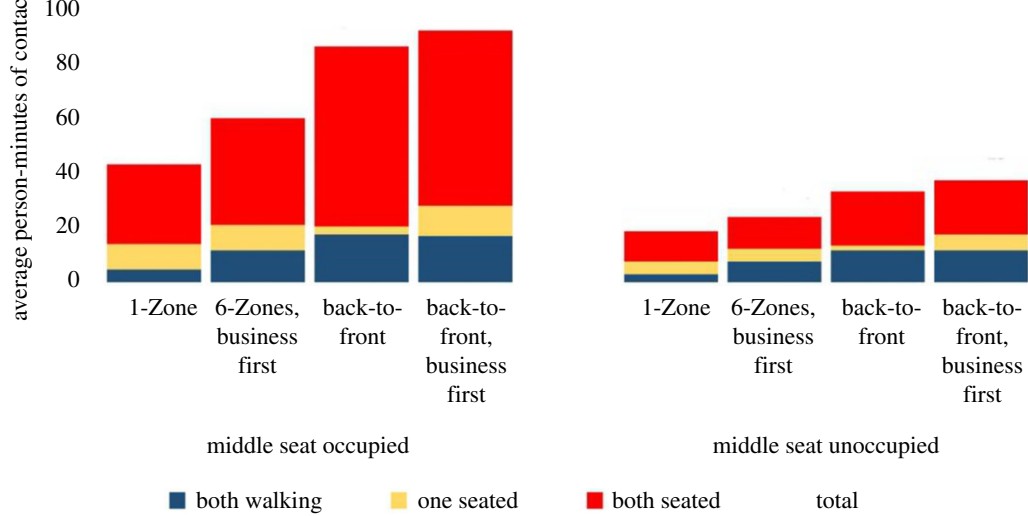

**Figure 10.** Contact mechanisms with overhead bin use permitted and a 1.818 m contact threshold without seat conflict.

differences between the boarding procedures decrease due to (i) a lower boarding time reducing the exposure between two seated passengers and (ii) less clustering in the aisle, reducing one weakness of the back-to-front boarding procedures. In fact, when middle seats are unoccupied without seat conflicts, all four procedures lead to similar exposure. In all other cases, the four procedures show similar relative trends, although with a smaller magnitude of difference between them.

We next examine whether changing the contact threshold changes the relative trends. If one procedure were to yield poor results mostly due to contacts close to 1.818 m away, but it yielded low counts for close-range contacts, then it might be effective in reducing infection risk, because close contacts are at higher risk of infection spread. Furthermore, if passengers wear masks, then social proximity would arise at a smaller threshold. We consider a contact threshold of 0.606 m, corresponding to a 2-feet distance. We see from figure 14 that the same relative trends hold, with 1-Zone having even a slightly greater advantage with overhead bin use permitted. Much of this decrease in social proximity with a 0.606 m threshold arises from a reduction in contacts between two seated passengers. Note that with middle seats unoccupied, contacts between seated passengers are generated from the business class alone, because the separation between the window and aisle seats

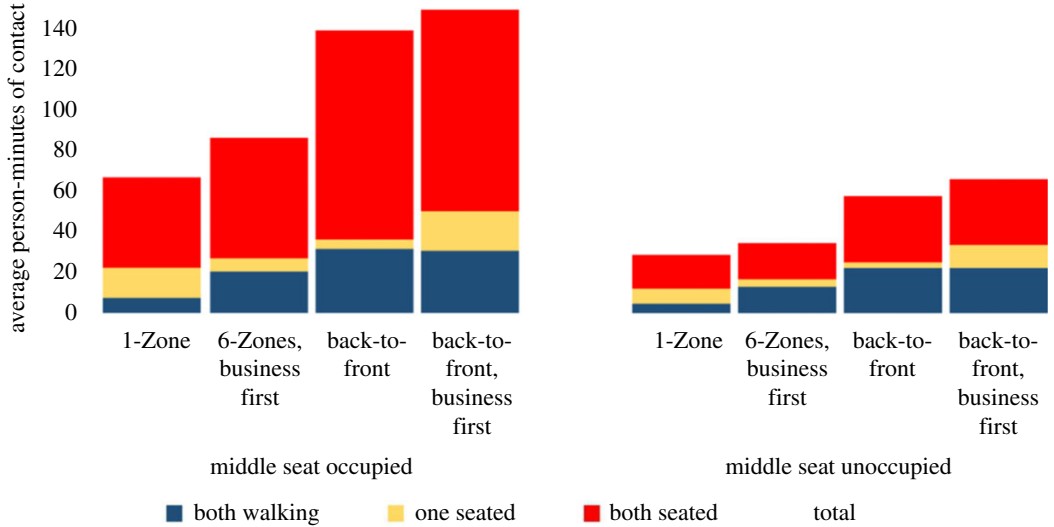

**Figure 11.** Contact mechanisms with overhead bin use permitted and a 1.818 m contact threshold with seat conflict.

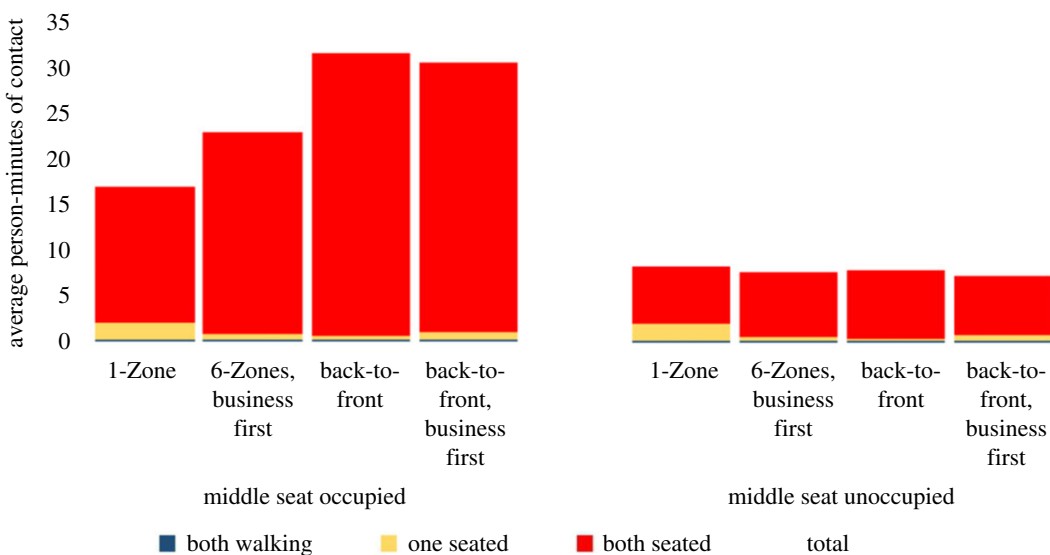

**Figure 12.** Contact mechanisms with overhead bin use prohibited and a 1.818 m contact threshold without seat conflict.

in economy class is higher than 0.606 m. The relative trends of the different procedures still hold because fewer zones produce substantially lower contacts between two passengers in the aisle. The net impact is that both with and without middle seats occupied, the relative differences between procedures remain roughly the same.

We finally examine users' exposures at the high end of total exposure to see if the four boarding procedures have the same relative trends as with average exposure. This analysis might provide insight, for example, for deterministic models that compute infection risk based on an exposure threshold. For each boarding procedure, we consider the top 10% of users by total exposure over the parameter sweep in figure 15. We see that here, again, the back-to-front boarding procedures fare much worse.

We next discuss some limitations of this work. First, we consider social proximity only during boarding. The infection could spread during the flight too, which we do not study. There are mixed results in the literature regarding the vulnerabilities during flights. Computational fluid dynamics simulations suggest that there could be a considerable effect of the cabin airflow [27]. On the other hand, a model based on empirical measurement of virus particles and passenger behaviour suggested that the risk is minimal and that other aspects of air travel, including boarding, could be responsible for outbreaks during air travel [28].

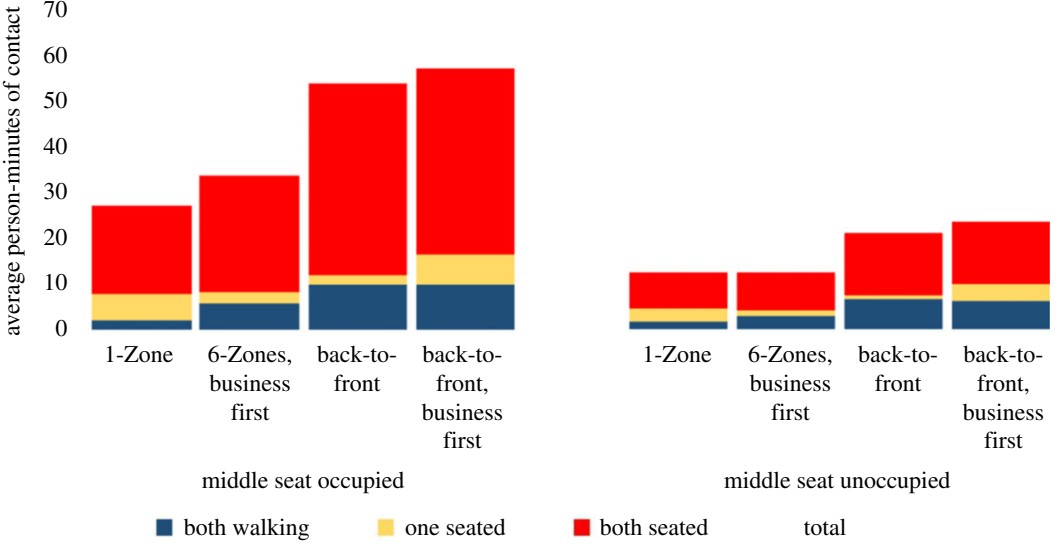

**Figure 13.** Contact mechanisms with overhead bin use prohibited and a 1.818 m contact threshold with seat conflict.

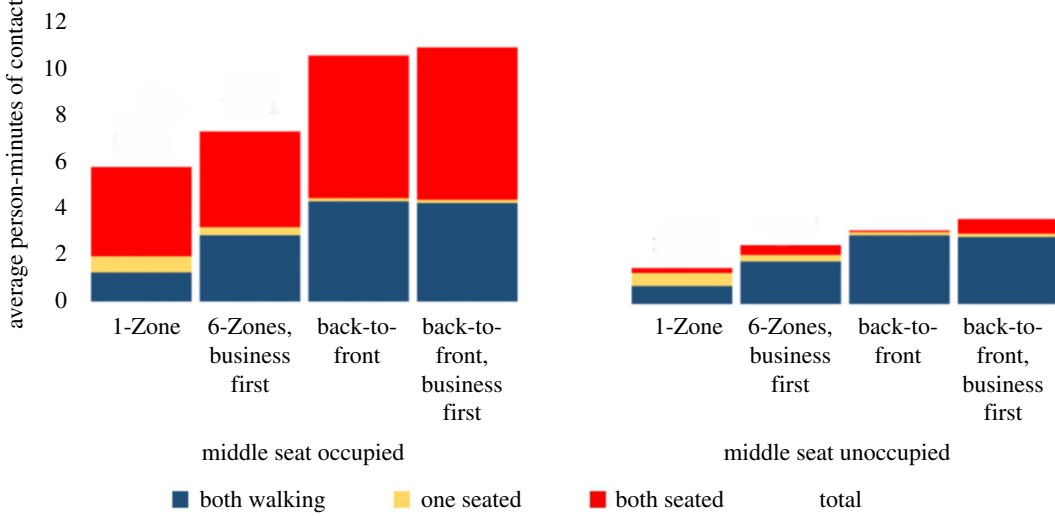

**Figure 14.** Contact mechanisms with overhead bin use permitted and a 0.606 m contact threshold without seat conflict.

Evidence from SARS outbreaks in aeroplanes suggests that both views have some merit. Out of 40 flights with SARS-infected persons on board, only five resulted in infections to passengers, with a total of 37 infected persons across these flights [4]. A single flight accounted for the majority of the infections. Passengers in rows close to the index passenger were disproportionately affected. However, half the infections occurred amongst passengers farther than two rows away. Given that inflight transmission risk is primarily to persons within two rows [4], the above observation suggests that boarding could play a significant role. In any case, given that the new boarding procedures are meant to reduce exposure to the virus during boarding, we focus on the merits of these changes alone in this paper.

Another limitation is that we estimate social proximity, rather than predict infection likelihood. The virus dose required for COVID-19 infection is not known, and we believe that quantitatively accurate predictions are not feasible at this point. Consequently, we focus on the relative benefits of different boarding procedures through the estimation of social proximity generated by them. Another limitation of using social proximity is that it does not account for direction-dependence of exposure. For example, someone facing another person who coughs may be at greater risk than someone at the same distance who is not facing the cougher. Such an analysis would need to consider differences arising from different sources or risk, such as coughing or breathing and airflow patterns, which are

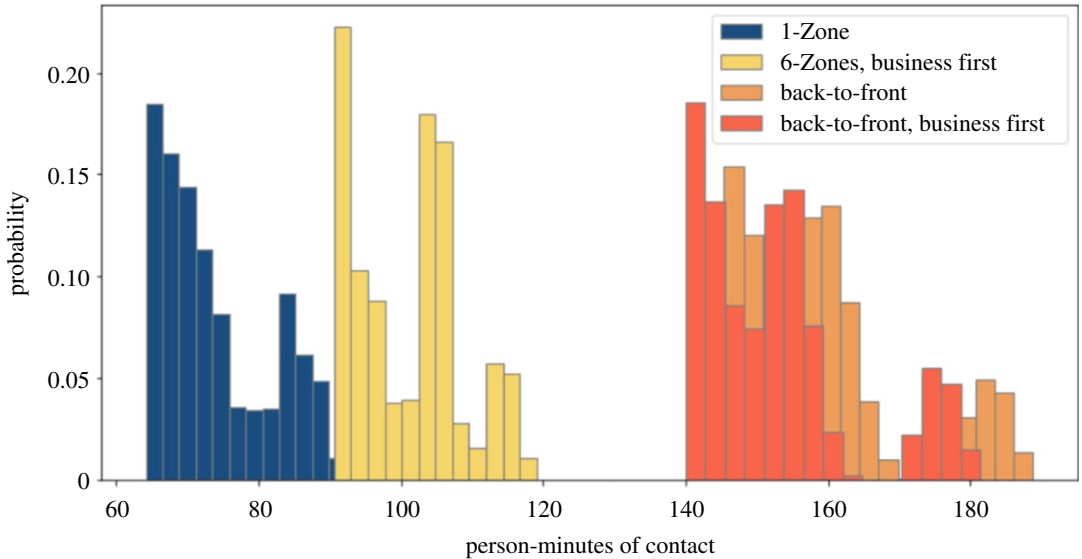

**Figure 15.** Exposure of individual users in the top 10% with overhead bin use permitted, middle seats occupied, and a 1.818 m contact threshold without seat conflict.

complex. Consequently, the directionality of contacts is generally not considered in either spatial network models [29] or agent-based epidemic models [30].

# 5. Conclusion

Our results show that while the original boarding procedures were substantially worse than they could have been, the new boarding procedures further worsen infection risk by increasing social proximity. Prohibiting the use of overhead bins to stow luggage, and boarding the window seat before the aisle seat, can ameliorate this. However, the new policies do not improve on the old ones in any situation studied here. We also identify the mechanisms leading to social proximity. Science-based changes to boarding procedures can decrease COVID-19 infection risk substantially by increasing social distance. In future work, we are working on a close-range infection transmission model, which will enable the estimation of actual infection risk.

Data accessibility. Data are available at the Dryad Digital Repository: https://doi.org/10.5061/dryad.18931zctb [31].

Authors' contributions. Conceptualization, methodology and validation: A.S. and S.N.; analysis and software development: T.I. and M.S.L.; visualization: S.N. and T.I.; insight on infection outbreaks: A.M. and M.S.; supervision: A.S.; writing: A.S.; reviewing and editing: all authors; funding acquisition: A.S., S.N., M.S. and A.M.

Competing interests. We declare we have no competing interests.

Funding. This material is based upon work supported by the National Science Foundation under grant nos. 1931511, 2027514, 2027529 and 2027518. Any opinions, findings and conclusions or recommendations expressed in this material are those of the authors and do not necessarily reflect the views of the National Science Foundation.

Acknowledgements. We thank an anonymous referee for suggesting the analysis of a ban on the use of overhead bins. This research used resources of the Argonne Leadership Computing Facility, which is a DOE Office of Science User Facility supported under Contract DE-AC02-06CH11357.

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
