## [Peer Review File · Royal Society Open Science]

Review History

RSOS-201019.R0 (Original submission)

Review form: Reviewer 1

Is the manuscript scientifically sound in its present form?

Yes

Are the interpretations and conclusions justified by the results?

No

Is the language acceptable?

Yes

Do you have any ethical concerns with this paper?

No

Have you any concerns about statistical analyses in this paper?

No

Recommendation?

Major revision is needed (please make suggestions in comments)

Comments to the Author(s)

The authors implemented an agent-based model to simulate different strategies for airplanes boarding dynamics, in order to assess SARS-COV-2 transmission potential during the boarding process.

The modeling framework is sound as it is mainly taken from existing literature. However, I have few major concerns about the paper results.

Major points:

1. The authors conclusions are supported by Figures 2 and 3, where the distribution of time for contacts within 1.82m are shown. However, my understanding is that the authors are cumulating every second of interacting individuals within such threshold, without considering that epidemiologically relevant contacts for SARS-COV-2 potential infection are considered to be those holding for more than 15 minutes [CDC, WHO]. Thus, I wonder if the reported differences in the four presented boarding strategies are still valid when only epidemiologically relevant contacts are considered. In principle, different boarding procedures might have different impact on the spreading potential once that only the relevant contacts are retained, or, on the contrary might show similar patterns with no potential benefits or drawbacks.

2. The model proposed by the authors requires several parameters. Unfortunately few (none?) of them seems to be informed by data, and some arbitrary choices are not further explored and discussed with a thorough sensitivity analysis (e.g. desired speed of passenger, intersection-speed/distance coefficients, luggage stowing time and distribution). Thus, it is hard to understand if the reported results are specific for the parameter choices or they are robust for different scenarios.

Minor points:

3. Citations have punctuation inconsistencies.

References:

[CDC] <https://www.cdc.gov/coronavirus/2019-ncov/global-covid-19/operational-considerations-contact-tracing.html>

[WHO] <https://apps.who.int/iris/rest/bitstreams/1277571/retrieve>

Review form: Reviewer 2

Is the manuscript scientifically sound in its present form?

Yes

Are the interpretations and conclusions justified by the results?

Yes

Is the language acceptable?

Yes

Do you have any ethical concerns with this paper?

No

Have you any concerns about statistical analyses in this paper?

No

Recommendation?

Major revision is needed (please make suggestions in comments)

Comments to the Author(s)

This paper is provoking and interesting.

The results look correct overall, although there some subtleties that are unclear to me.

I have a few concerns:

- droplet emission is likely to be mostly related to talking (or coughing, heavy breathing) and directional. There could therefore be quite a strong difference in the risk of the different contributions discussed in the Discussion. In view of this, the policy claims in the abstract should be lessened.
- The rationale for a back-to-front is also that passengers seating in back seats would not be slowed down and forced in a long queue waiting for passengers in front seats to stow their luggage, find their seat etc. Since any oversimplified model would give this as a main factor, I think the authors should explain more clearly why their counterintuitive result holds.
- it is unclear how much of the result is caused by the luggage stowing time. Many airlines and regulators have forbidden to bring hand luggage, precisely on the suspicion that this would be a major factor in the effect discussed by the authors. The authors should clarify if that is indeed a major factor, or not, by comparing also the case of no stowing. Naively, I suspect so, since in their model the stowing time is of the same order of magnitude as the minimum time spent walking towards their seat, and slows down other passengers, increasing contacts in the aisle.
- A comment on low-cost airlines. Companies with less seat space have the additional problem of passengers having to move back to the aisle to allow others to seat near windows. Would a window/aisle strategy work better in that case?

Decision letter (RSOS-201019.R0)

Dear Dr Srinivasan

The Editors assigned to your paper RSOS-201019 "From Bad to Worse: Airline Boarding Changes in Response to COVID-19" have now received comments from reviewers and would like you to revise the paper in accordance with the reviewer comments and any comments from the Editors. Please note this decision does not guarantee eventual acceptance.

Please submit your revised manuscript and required files (see below) no later than 21 days from today's (ie 30-Nov-2020) date. Note: the ScholarOne system will 'lock' if submission of the revision is attempted 21 or more days after the deadline. If you do not think you will be able to meet this deadline please contact the editorial office immediately.

on behalf of Dr Mirco Musolesi (Associate Editor) and Marta Kwiatkowska (Subject Editor)
openscience@royalsociety.org

Associate Editor Comments to Author (Dr Mirco Musolesi):

Associate Editor: 1

Comments to the Author:

The reviewers raised several concerns that need to be carefully addressed by the authors. For this reason, I would recommend a major revision of this manuscript.

Reviewer comments to Author:

Reviewer: 1

Comments to the Author(s)

The authors implemented an agent-based model to simulate different strategies for airplanes boarding dynamics, in order to assess SARS-COV-2 transmission potential during the boarding process.

The modeling framework is sound as it is mainly taken from existing literature. However, I have few major concerns about the paper results.

Major points:

1. The authors conclusions are supported by Figures 2 and 3, where the distribution of time for contacts within 1.82m are shown. However, my understanding is that the authors are cumulating every second of interacting individuals within such threshold, without considering that epidemiologically relevant contacts for SARS-COV-2 potential infection are considered to be those holding for more than 15 minutes [CDC, WHO]. Thus, I wonder if the reported differences in the four presented boarding strategies are still valid when only epidemiologically relevant contacts are considered. In principle, different boarding procedures might have different impact

on the spreading potential once that only the relevant contacts are retained, or, on the contrary might show similar patterns with no potential benefits or drawbacks.

2. The model proposed by the authors requires several parameters. Unfortunately few (none?) of them seems to be informed by data, and some arbitrary choices are not further explored and discussed with a thorough sensitivity analysis (e.g. desired speed of passenger, intersection-speed/distance coefficients, luggage stowing time and distribution). Thus, it is hard to understand if the reported results are specific for the parameter choices or they are robust for different scenarios.

Minor points:

3. Citations have punctuation inconsistencies.

References:

[CDC] <https://www.cdc.gov/coronavirus/2019-ncov/global-covid-19/operational-considerations-contact-tracing.html>

[WHO] <https://apps.who.int/iris/rest/bitstreams/1277571/retrieve>

Reviewer: 2

Comments to the Author(s)

This paper is provoking and interesting.

The results look correct overall, although there some subtleties that are unclear to me.

I have a few concerns:

- droplet emission is likely to be mostly related to talking (or coughing, heavy breathing) and directional. There could therefore be quite a strong difference in the risk of the different contributions discussed in the Discussion. In view of this, the policy claims in the abstract should be lessened.

- The rationale for a back-to-front is also that passengers seating in back seats would not be slowed down and forced in a long queue waiting for passengers in front seats to stow their luggage, find their seat etc. Since any oversimplified model would give this as a main factor, I think the authors should explain more clearly why their counterintuitive result holds.

- it is unclear how much of the result is caused by the luggage stowing time. Many airlines and regulators have forbidden to bring hand luggage, precisely on the suspicion that this would be a major factor in the effect discussed by the authors. The authors should clarify if that is indeed a major factor, or not, by comparing also the case of no stowing. Naively, I suspect so, since in their model the stowing time is of the same order of magnitude as the minimum time spent walking towards their seat, and slows down other passengers, increasing contacts in the aisle.

- A comment on low-cost airlines. Companies with less seat space have the additional problem of passengers having to move back to the aisle to allow others to seat near windows. Would a window/aisle strategy work better in that case?

===PREPARING YOUR MANUSCRIPT===

a 'clean' version of the new manuscript that incorporates the changes made, but does not highlight them. This version will be used for typesetting if your manuscript is accepted. Please ensure that any equations included in the paper are editable text and not embedded images.

===PREPARING YOUR REVISION IN SCHOLARONE===

- If you are requesting a discretionary waiver for the article processing charge, the waiver form must be included at this step.
- If you are providing image files for potential cover images, please upload these at this step, and inform the editorial office you have done so. You must hold the copyright to any image provided.
- A copy of your point-by-point response to referees and Editors. This will expedite the preparation of your proof.

- Ensure that your data access statement meets the requirements at <https://royalsociety.org/journals/authors/author-guidelines/#data>. You should ensure that you cite the dataset in your reference list. If you have deposited data etc in the Dryad repository, please include both the 'For publication' link and 'For review' link at this stage.
- If you are requesting an article processing charge waiver, you must select the relevant waiver option (if requesting a discretionary waiver, the form should have been uploaded at Step 3 'File upload' above).
- If you have uploaded ESM files, please ensure you follow the guidance at <https://royalsociety.org/journals/authors/author-guidelines/#supplementary-material> to include a suitable title and informative caption. An example of appropriate titling and captioning may be found at https://figshare.com/articles/Table_S2_from_Is_there_a_trade-off_between_peak_performance_and_performance_breadth_across_temperatures_for_aerobic_scop_e_in_teleost_fishes_/3843624.

Author's Response to Decision Letter for (RSOS-201019.R0)

See Appendix A.

RSOS-201019.R1 (Revision)

Review form: Reviewer 1

Is the manuscript scientifically sound in its present form?

Yes

Are the interpretations and conclusions justified by the results?

Yes

Is the language acceptable?

Yes

Do you have any ethical concerns with this paper?

No

Have you any concerns about statistical analyses in this paper?

No

Recommendation?

Accept as is

Comments to the Author(s)

The authors promptly addressed all the issues raised in the previous round of review and further extend the manuscript. The discussion of different scenarios and modeling choices is definitely improved.

Decision letter (RSOS-201019.R1)

Dear Dr Srinivasan,

It is a pleasure to accept your manuscript entitled "From Bad to Worse: Airline Boarding Changes in Response to COVID-19" in its current form for publication in Royal Society Open Science. The comments of the reviewer(s) who reviewed your manuscript are included at the foot of this letter.

COVID-19 rapid publication process:

We are taking steps to expedite the publication of research relevant to the pandemic. If you wish, you can opt to have your paper published as soon as it is ready, rather than waiting for it to be published the scheduled Wednesday.

This means your paper will not be included in the weekly media round-up which the Society sends to journalists ahead of publication. However, it will still appear in the COVID-19 Publishing Collection which journalists will be directed to each week (<https://royalsocietypublishing.org/topic/special-collections/novel-coronavirus-outbreak>).

If you wish to have your paper considered for immediate publication, or to discuss further, please notify openscience_proofs@royalsociety.org and press@royalsociety.org when you respond to this email.

on behalf of Dr Mirco Musolesi (Associate Editor) and Marta Kwiatkowska (Subject Editor)
openscience@royalsociety.org

Associate Editor Comments to Author (Dr Mirco Musolesi):

Comments to the Author:

The reviewer suggested to accept the article in this new revised form (as it is) and I agree with this assessment.

Reviewer comments to Author:

Reviewer: 1

Comments to the Author(s)

The authors promptly addressed all the issues raised in the previous round of review and further extend the manuscript. The discussion of different scenarios and modeling choices is definitely improved.

Appendix A

Response to Reviewers

We thank the reviewers for their feedback. We have made changes to clarify possible misunderstandings and discussed the impact of hand luggage policies, as suggested by a reviewer. We summarize the changes below and provide other clarifications to address reviewer comments.

Reviewer 1

1. The authors conclusions are supported by Figures 2 and 3, where the distribution of time for contacts within 1.82m are shown. However, my understanding is that the authors are cumulating every second of interacting individuals within such threshold, without considering that epidemiologically relevant contacts for SARS-COV-2 potential infection are considered to be those holding for more than 15 minutes [CDC, WHO]. Thus, I wonder if the reported differences in the four presented boarding strategies are still valid when only epidemiologically relevant contacts are considered. In principle, different boarding procedures might have different impact on the spreading potential once that only the relevant contacts are retained, or, on the contrary might show similar patterns with no potential benefits or drawbacks.

Response: The reviewer is correct about how we calculate the exposure. Our approach is consistent with models that use a dose-response relationship to model infection risk. The citations provided by the reviewer define contacts for contact tracing purposes rather than for estimating infection risk. We have now explained why these two differ in Section 4.1. We have added citations that (i) clarify that use of contact tracing metrics for infection spread estimate would miss infections, (ii) show that COVID-19 spread occurs with much less than 15-minute exposure, and (iii) demonstrate infection risk modeling based on total exposure time. We have also added a histogram of exposure for individual users (Fig. 15) so that readers interested in only large exposures will find that information. This figure too shows the disadvantage of the back-to-front process.

2. The model proposed by the authors requires several parameters. Unfortunately few (none?) of them seems to be informed by data, and some arbitrary choices are not further explored and discussed with a thorough sensitivity analysis (e.g. desired speed of passenger, intersection-speed/distance coefficients, luggage stowing time and distribution). Thus, it is hard to understand if the reported results are specific for the parameter choices or they are robust for different scenarios.

Response: The choice of parameter values is from the original CALM model paper [15]. They are either physical parameters, such as human movement speed, that are from literature and explained in [15], or parameters for the CALM model that have been validated against empirical data in [15]. We had earlier cited this paper only for the parameters mention in Section 3. We have now cited it for the parameters in Section 4 too. We have also now explained that we use a subset of the range in the above paper for the intersection-distance-

threshold parameter. The reason is that the above paper simulated four airplanes, ranging from 50 seats to over 200 seats. The current paper deals with only one airplane and does not need to consider the lowest or highest end of the range. We have also added a seat conflict time parameter to study an issue suggested by another reviewer. We have added a reference to justify the parameter value for this, based on the minimum value in [29] and an additional range to account for uncertainty as used in [29].

We also wish to clarify the results presented are those for a parameter sweep that generates 16,000 scenarios using a variety of parameter values. We had mentioned these in Section 4.1. We have now also clarified this at the beginning of Section 4.2.

3. Citations have punctuation inconsistencies.

Response: We have corrected these.

Reviewer 2

4. Droplet emission is likely to be mostly related to talking (or coughing, heavy breathing) and directional. There could therefore be quite a strong difference in the risk of the different contributions discussed in the Discussion. In view of this, the policy claims in the abstract should be lessened.

Response: We understand the reviewer's point. We have changed the last sentence of the abstract to summarize the knowledge produced from the paper rather than make a policy recommendation. We also include this in the description of the limitations of this work in Section 5. However, we also point out that it is quite common for models not to take direction into account, and cite an example.

5. The rationale for a back-to-front is also that passengers seating in back seats would not be slowed down and forced in a long queue waiting for passengers in front seats to stow their luggage, find their seat etc. Since any oversimplified model would give this as a main factor, I think the authors should explain more clearly why their counterintuitive result holds.

Response: Prior results and also the rationale given by airlines point out to a different intuition. (i) We had mentioned in Section 2 that our prior results [10] suggest the intuition demonstrated by our current results. We had shown in previous work that random boarding leads to an average lower time waiting for others to stow their luggage than if one had several zones. The former leads to several short queues spread throughout the plane while the latter brings more people together in the same place. Due to the roughly quadratic relationship between the number of people and interactions, it is preferable to have many small queues rather than a few large ones. (ii) We had provided a citation for the intuition behind the new policy in Section 2

[7]. It is meant to reduce the contact between seated passengers and others who walk past them. But our results show that such contact is for a very short duration and thus not the dominant factor in an exposure. This observation is consistent with the intuition based on our prior published results.

6. It is unclear how much of the result is caused by the luggage stowing time. Many airlines and regulators have forbidden to bring hand luggage, precisely on the suspicion that this would be a major factor in the effect discussed by the authors. The authors should clarify if that is indeed a major factor, or not, by comparing also the case of no stowing. Naively, I suspect so, since in their model the stowing time is of the same order of magnitude as the minimum time spent walking towards their seat, and slows down other passengers, increasing contacts in the aisle.

Response: This is an interesting observation. We discuss this along with the next comment because the total time for stowing the luggage and waiting for seated passengers to give way impacts the wait for others in the aisle. We have obtained insights from additional simulations that we have now reported. We note that the USA currently lacks regulations forbidding hand luggage. Our results suggest that such a rule, in addition to the one indicated by the reviewer in comment #7, would reduce the exposure risk substantially.

7. A comment on low-cost airlines. Companies with less seat space have the additional problem of passengers having to move back to the aisle to allow others to seat near windows. Would a window/aisle strategy work better in that case?

Response: We have performed additional simulations to examine the following new policies in addition to the prior one (use of overhead bins permitted and a window to aisle boarding strategy): (i) no use of overhead bins for stowing hand luggage and a window to aisle strategy in boarding, (ii) use of overhead bins and no window to aisle strategy in boarding, and (iii) no use of overhead bins and no window to aisle strategy in boarding. Our results show that strategy (i) leads to all boarding options perform similarly when the middle seats are unoccupied. One contribution toward this arises from a reduction in time waiting for other passengers to be seated (including time for stowing their luggage). Another stems from the fact that the overall boarding process is faster with this option. So, the exposure between two seated passengers is reduced, which benefits the back-to-front strategy particularly. All other strategies show back-to-front boarding as substantially worse than the alternatives.

In summary, random boarding is significantly better than other strategies. When the four basic boarding strategies are not combined with a ban on the use of overhead bins, the use of a window to aisle boarding order, and keeping middle seats unoccupied, the back-to-front strategies are substantially worse.